# Investigating the diverse potential of a multi-purpose legume, *Lablab purpureus* (L.) Sweet, for smallholder production in East Africa

**Alison Nord**[1]*, **Neil R. Miller**[2], **Wilfred Mariki**[3], **Laurie Drinkwater**[4], **Sieglinde Snapp**[1]

**1** Plant, Soil and Microbial Sciences Department, Michigan State University, East Lansing, Michigan, United States of America, **2** Canadian Foodgrains Bank, Arusha, Tanzania, **3** Tanzania Agriculture Research Institute – Selian Centre, Arusha, Tanzania, **4** School of Integrative Plant Science, Horticulture, Cornell University, Ithaca, New York, United States of America

* nordalis@msu.edu

**Data Availability Statement:** All data files are available from the Harvard Dataverse SIIL Tanzania database (doi:10.7910/DVN/MKIUUB).

## Abstract

Climate change is posing severe challenges in Africa, where resilient crops are urgently needed to withstand drought periods and unreliable rainfall. Multi-purpose legume species, such as lablab (*Lablab purpureus* (L.) Sweet), have been under-utilized yet have the potential to overcome climate challenges. While lablab is native to Africa, there are few characterized varieties and it is under-utilized by smallholder farmers due to a lack of information and access to varieties. Knowledge is especially lacking on the performance of this crop by genotype, management, and environment. We conducted a two-year field study at two sites to evaluate 29 lablab cultivars under sole and maize intercrop management, with 14 cultivars selected for in-depth study. Cultivars were evaluated on vegetative biomass and grain yield production, with N fixation assessed for one site year. Biomass and grain production differed across environments and cultivars, with only biomass affected by intercropping. Average grain yield was substantially reduced to only 37 kg ha$^{-1}$ in environments with maximum temperatures greater than 33°C, but biomass production yielded comparable amounts across high temperatures and in dry (<500 mm rainfall) environments. Tradeoffs were found between biomass and grain yield across high yielding cultivars, with the top three grain accessions averaging 612 kg ha$^{-1}$ of grain and 1.97 Mg ha$^{-1}$ biomass whereas the top three biomass accessions produced 327 kg ha$^{-1}$ grain and 2.52 Mg ha$^{-1}$ biomass across all environments. In a comparison of production and N fixation measurements, cultivars were identified which may have high performance in both. Suitability of lablab for grain and biomass production were visualized across Tanzania in a map comparing max temperature thresholds for grain and biomass against average regional livestock populations. This provides a way forward for identifying potential areas for lablab cultivation as a novel means to enhance fodder and pulse production with smallholder farmers.

**Funding:** This study was made possible by the support of the American People provided to the Feed the Future Innovation Lab for Sustainable Intensification through the United States Agency for International Development (USAID)(https://www.usaid.gov). The contents are the sole responsibility of the authors and do not necessarily reflect the views of USAID or the United States Government. Program activities are funded by the United States Agency for International Development (USAID) under Cooperative Agreement No. AID-OAA-L-14-00006 received by authors S.S., A.N, N.M. The funders had no role in study design, data collection and analysis, decision to publish, or preparation of the manuscript.

**Competing interests:** The authors have declared no competing interests exist.

## Introduction

As African populations exponentially increase under current food production dominated by smallholder agriculture, intensification without further degrading the natural resources of these systems is greatly needed [1]. Legume intensification is one pathway through which smallholder farmer production may be sustainably increased and involves complementing a farmer's current cropping system through incorporating legumes [2]. Smallholder farmers have historically grown legumes in ways that have complemented cereal and cash crops, such as maize bean rotations and cereal-legume intercrops [3]. Increasing this legume presence in maize cropping systems provides many potential benefits such as increased crop diversity, improved soil fertility without the overuse of chemical inputs, increased household dietary diversity, and increased cash income through sale of high market-value legumes [1]. Despite these benefits, maize monocrops have continued to be promoted to smallholder farmers, at the expense of legume production. Challenges to increasing legume production, such as limited availability of legume seed, pest problems, lack of markets, and low yields, have held farmers back from adopting more legume intensive systems, and must be addressed for sustainable intensification to occur.

Challenges to legume production have been exacerbated by the limited nature of legume research and minimization of the multi-purpose nature of legumes. Legume studies often prioritize either the grain or forage potential of the study crops, with less focus on the tradeoffs or interactions of these traits [4,5]. Studies on farmer objectives in growing legumes however confirm that farmers have multiple production objectives in growing legumes and consider other benefits besides just grain yield in choosing to cultivate legumes, such as improving soil fertility [2,6,7]. A legume's ability to improve soil fertility through nitrogen (N) fixation depends on many different factors, including genetics, management and the environment [3]. N contribution by a legume therefore must be considered across these factors to fully understand the legume's effect within a system. The diverse objectives of farmers must be considered in order for appropriate legumes to be identified that will meet farmer production needs and improve farming systems. As such, studies should evaluate legume potential from multiple angles, such as grain, forage, and N fixation potential, across diverse cultivars and environments for a more robust assessment of these crops.

Typical legume studies, such as those for common bean and cowpea, focus on sole cropping and singular productivity measurements in assessing crop potential and potential across cultivars [8,9,10,11]. There is a lack of quantifying multiple production traits and understanding tradeoffs of these traits within cultivars and in systems that resemble local farmer context, such as intercropped with maize. Many cultivar studies have instead focused on finding a few top grain producing types that fit across environments [12]. However, identifying appropriate legume cultivars that fit within different farming systems requires testing diverse cultivars for multiple production qualities and testing their performance under different environment and management conditions.

Overall there is a need for better understanding of environment and management parameters of legumes, especially those with multipurpose traits. Previous legume studies have been too empirical and fail to look at legume management as a system within which growing parameters may be established [13]. This is especially true for understudied multipurpose legumes, and there are few systematic studies that identify ways of introducing novel crops. Lablab (*Lablab purpureus* (L.) Sweet) is one such understudied legume with limited study of its diverse genetic collection and evaluation of its multipurpose qualities [14]. Our study takes a multi-dimensional approach to assessing lablab amongst different genetic sources, environments, and management, across which these effects are not well understood. Our overall objective was

to identify promising lablab accessions and suitable growing conditions to inform lablab integration into smallholder farming systems. Specifically, we aimed to identify lablab accessions that are high yielding and stable across environments as well as those that perform best in terms of grain yield and biomass in specific environments and sole cropped or intercropped with maize. We further wanted to assess accession performance across vegetative biomass, N fixation, and grain yield to determine whether some accessions have high multipurpose potential or if accessions are more likely to perform well in one trait over another.

## Materials and methods

### Study sites

The study was conducted over the 2016 and 2017 growing seasons at two sites in the Northern Zone of Tanzania, one at the Tanzania Agricultural Research Institute Selian Centre (SARI) located in Arusha and the other at the Tropical Pesticides Research Institute (TPRI) research farm in Miwaleni, Moshi. The SARI site is at an altitude of 1387 MASL with a mean annual rainfall of 1052 mm and mean annual temperature of 19.5 ˚C. The TPRI site represents lowland areas at 719 MASL with a mean annual rainfall of 600 mm and mean annual temperature of 23.5˚C. Both sites are weakly bimodal, with the majority of rainfall occurring in March through May and a short rain period between November and January. Most field crops are planted in the longer rain period of March—May whereas few crops are planted in the unreliable rains that occur November—January.

### Experimental design

The experimental design of the field trials included two factors, accession and cropping system, arranged in a modified split plot with three blocks replicated at each site. The accessions consisted of 29 lablab accessions and 3 cowpea varieties chosen as a reference crop. The lablab accessions included a selection from a core collection identified by Pengelly and Maass [15] with 5 varieties registered in Kenya and landraces collected throughout East Africa. This study focuses on measurements from 14 lablab accessions chosen as a subset of the 29 total accessions studied with one of the cowpea varieties chosen for reference. Description of the full 29 accessions can be found in Miller et al. [16] with preliminary performance assessment. The 14-accession subset was chosen based on those that had shown promise from early observations of the full set of accessions, with the goal of selecting cultivars with a range of growth types (Table 1). Four of these accessions were subsequently chosen for further study through on-farm trials with the purpose of selecting a final set of accessions for registration. The cropping system factor consisted of each lablab accession sole-cropped or intercropped with maize (Pannar 15). To simplify field operations, cropping system was randomly arranged within blocks in strips of consecutive intercropped or sole-cropped plots. Each strip had either 8 plots (SARI) or 7 (TPRI). One sole maize plot was also included in each block. Individual plots were 4.5 by 5.4 m with 1.5 m unplanted borders between plots within strips. Lablab spacing was 0.9 m between rows and 0.5 m within rows with five rows per plot and two seeds planted per station (4.4 seeds m$^{-2}$). Cowpea spacing was half that of lablab, with 0.45 m between rows and 0.5 m within rows. Planting was done in an additive design, where lablab and cowpea spacing was the same intercropped with maize as it was sole cropped. Maize was planted between rows with lablab or cowpea, with six rows per plot at 0.9 m between rows and 0.5 m within rows and two seeds planted per station for a seeding rate of 4.4 seeds m$^{-2}$. One maize row was planted at the borders of all sole cropped plots to ensure uniform shading regardless of whether sole-cropped plots were adjacent to intercropped plots.

**Table 1. Lablab accessions and cowpea reference variety used in the study.**

| No. | Accession | Maturity | Seed color | Seed Wt. (g/100 seeds) | Qualities | Origin | Other Properties |
|---|---|---|---|---|---|---|---|
| 1 | CIAT 22759 | Early-mid | Black | 30 | Indeterminate | Kenya | Forage type |
| 3 | DL1001 | Late | Brown | 23 | Indeterminate | Kenya | Dual purpose |
| 4[†] | DL1002 | Early | Black | 26 | Semi-determinate | Kenya | Popular landrace |
| 6[†] | Echo Cream | Mid | White | 30 | Indeterminate | Tanzania | |
| 8 | Highworth | Early | Black | 25 | | India | Forage variety, Popular forage variety |
| 12 | ILRI 13700 | Very late | Black | 38 | Vigorous growth | Ethiopia | |
| 14 | ILRI 14437 | Early-mid | Black | 23 | | Unknown | |
| 16 | ILRI 6930 | Early-mid | Brown | 31 | Long pods, high biomass | Unknown | Drought tolerant |
| 17[†] | Karamoja Red | Mid | Red | 36 | | Uganda | |
| 21 | PI 195851 | Very late | Dark brown | 23 | High biomass, low grain | Egypt | Drought tolerant |
| 22[†] | Q 6880B | Very early | Black | 22 | Short-season | Brazil | Dual purpose |
| 23 | Rongai | Very late | Tan | 26 | Indeterminate | Kenya | Popular forage variety |
| 25 | SARI Nyeupe | Late | White | 28 | | Tanzania | |
| 26 | SARI Rongai | Mid | Black | 30 | | Tanzania | |
| 31 | Fadhari cowpea | Mid-late | Red | 11 | Spreading growth | Tanzania | |

[†]Accessions chosen for continuation to on-farm trials

## Management

The first trial was established in March of 2016, with field preparation and plantings occurring in early March at the SARI site. Maize was planted first at SARI on 11 March 2016, with lablab seeded 12 days later. The TPRI site was started later, with maize planted on 6 April 2016 and lablab seeded 8 days later. In 2017, maize was planted at the SARI site on 17 March 2017 and lablab seeded 3 days later. Maize planting at TPRI site started earlier on 8 March 2017 and lablab seeded 6 days later. Across all sites and years DAP fertilizer (18-46-0) was applied to maize at planting with a rate of 77 kg ha$^{-1}$. Urea (46-0-0) was side dressed at 110 kg ha$^{-1}$. Fields were tilled with a disc plow in 2016 but planted without tillage in 2017. Weed control was achieved using a pre-plant glyphosate application (2.5 L ha$^{-1}$) at planting, and by hand-hoe throughout the growing season as needed. Insecticide was applied as needed at both sites as significant insect pest damage was observed.

## Plant and soil measurements

Above-ground biomass was sampled for the lablab subset previously identified to quantify biomass yields and sample tissue for $^{15}$N analysis. Destructive sampling of plants was done during the early podding growth stage. In 2016 this occurred at SARI 98 and 112 days after planting (DAP) and 98 DAP at TPRI. In 2017 biomass was 103–107 DAP at SARI and 77 DAP at TPRI. Plants were sampled within a 0.9 m by 3 m sampling frame in 2016 and a 0.9 m by 2 m sampling frame in 2017. Fresh weight of lablab was measured in the field, and sub samples were taken for dry weight and further sampling.

Maize grain yield was determined from a sampling frame of 3 m across 2 rows (2016) and 3 rows (2017) within each plot. Grain was air dried then weighed and moisture content of the grain recorded. Due to the range of maturity between lablab accessions, and the indeterminate nature of most accessions, lablab grain harvest began as soon as dried pods were present and continued across several months during which most plots were harvested multiple times. In 2016 this occurred over five harvest dates at SARI and two at TPRI. In 2017 SARI lablab harvest had four

harvest dates and TPRI harvest occurred over three dates. Lablab pods were hand-harvested using a 3 m x 4.5 m sampling frame. In addition to weighing dry pod weight at each harvest date, pods from all harvest dates were combined to be threshed and weighed for determination of plot yield.

Soil samples were taken from each site for baseline soil properties at 0–20 cm and 20–40 cm depths, presented in supporting information S1 Table. A composite soil sample was collected for each block and analyzed for texture, pH, EC, and P.

**Nitrogen sub study—SARI 2017.** In 2017, root and nodule biomass were recorded from the SARI site from the sole cropped plots of the lablab subset. Roots were sampled from three locations per plot using a soil corer (4,415 cm$^3$) centered over a lablab plant. All nodules from the root samples were counted with color recorded to determine effectiveness and weighed after drying. Above-ground biomass sub-samples were oven dried at 70˚C, ground in a Wiley mill through a 0.5 mm then finely ground in a ball mill in preparation for $^{15}$N analysis. Samples were sent to the University of California Davis Stable Isotope Facility, CA, USA for $^{15}$N analysis using a PDZ Europa 20–20 isotope ratio mass spectrometer. The resulting $^{15}$N natural abundance of the samples was calculated using the equation: $\delta^{15}N(‰) = 1000 \left[ \left( \frac{R_{sample}}{R_{standard}} \right) - 1 \right]$ where $\delta^{15}N$ is expressed in parts per thousand (‰) and R is the ratio of $^{15}$N/$^{14}$N in the sample [17]. Additional soil samples were taken at the SARI site from the lablab subset plots following the 2016 and 2017 growing seasons to analyze soil nitrate and ammonium using a 2 M KCl extraction.

## Meteorological data

Rainfall data was collected at both locations for the 2016 and 2017 growing seasons. SARI rainfall measurements were reported by the Arusha Airport weather station located approximately 1.4 km away and TPRI rainfall measurements were obtained from a rain gauge located on site. Temperature data was retrieved through remote sensing from the Terra MODIS dataset provided by the USGS as day and night temperatures in 8-day increments at a 1 km resolution [18]. These temperatures were averaged per month across 2016 and 2017 and reported for the main growing season (January—September).

## Land equivalency ratio

The efficiency of the lablab-maize intercrop compared to sole cropping was evaluated using the land equivalency ratio (LER). LER is defined as

$$LER = \frac{Y_1}{M_1} + \frac{Y_2}{M_2}$$

where $Y_1$ and $Y_2$ are the intercrop yields of crop 1 and crop 2 and $M_1$ and $M_2$ are the sole cropped yields of crop 1 and crop 2. In this study $Y_1$ and $M_1$ were defined as maize grain yield intercropped and sole cropped respectively [19]. Given that lablab is often grown both for grain and fodder, two types of LERs were calculated to assess production of grain yield and fodder in intercrop systems with maize. The grain LER defined $Y_2$ as lablab grain yield intercropped and $M_2$ as sole cropped grain yield. The fodder LER defined $Y_2$ as lablab biomass intercropped and $M_2$ as sole cropped biomass. LER was calculated per accession by block within each environment and results are reported as average LER for each site year.

## Data analysis

Lablab biomass and grain yield were analyzed by a three-way analysis of variance (ANOVA) in SAS 9.7 using PROC MIXED to compare differences across environments (year by site),

accession, and management (intercrop vs. sole crop). The model included block nested within environment and management by block as random effects. Maize yield was analyzed by a two-way ANOVA to compare differences across environment and accession. In this model block nested within environment was set as a random effect. While the cowpea variety (#31) yields are presented for comparison, they were not included in statistical analyses with lablab and instead were analyzed separately for differences between environment and management.

A principal component analysis (PCA) was done using PROC PRINCOM in SAS to analyze the nitrogen sub study data collected from SARI 2017. This analysis generated variables representing crop productivity and nitrogen fixation within sole cropped lablab plots at SARI in 2017 and assessed multivariate accession effects. All data points for each of the 14 accessions across three blocks were used in the analysis. Variables included lablab grain yield, biomass, soil nitrate, nodule weight, $\delta^{15}N$, maturity (days to 50% flowering), plant population, and %N of biomass. Principal components with eigenvalues greater than 1 and accounting for more than 15% of the variability in the data were retained. Principal components 1 and 2 were further analyzed by a one-way ANOVA with block set as a random effect to compare accessions across components with the Tukey-Kramer test used to identify accession mean differences (alpha = 0.05).

Analysis of multivariate stability statistics was done with the accession main effect plus accession by environment interaction for grain yield and biomass using the GGEBiplotGUI package with RStudio in R statistical software. Two biplot views, "mean vs. stability" and "which-won-where" were used to visually assess accession performance across environments for grain yield and biomass as well as to determine tradeoffs among high performing accessions for both traits. These biplots have been identified as best capturing genotype by environment effects for multi-environment variety trials [20]. Accession measurements were averaged across management practices for each environment to obtain mean performance for each trait which was subjected to the GGE biplot analysis. The data in "Mean vs. Stability" view was not scaled (Scaling = 0), environment-centered (Centering = 2) and based on genotype-focused singular value partitioning (SVP = 1). The "which-won-where" model parameters were also set on un-scaled data (Scaling = 0), environment-centered (Centering = 2) and environment focused singular value partitioning (SVP = 2)[21].

## Results

### Weather

Rainfall across both study years was below average and unevenly distributed across months for both sites. In 2016 the SARI site had 315 mm rainfall between January and September, with the majority of rainfall occurring between January and April (Fig 1A). For the same time period the TPRI site had 252 mm of rainfall with the majority of rainfall occurring in April after the field trials were planted (Fig 1B). In 2017 the SARI site had 463 mm of rainfall with the majority occurring in April and May, later in the year than 2016. The TPRI site had 311 mm of rainfall in that same time period, with the majority of rainfall also occurring in April and May. Temperatures at SARI were consistent across the two years, with average maximum/minimum temperatures during the lablab growing period (March—September) of 29˚C/16˚C in 2016 and 28˚C/16˚C in 2017 (Fig 1). Average maximum/minimum temperature for TPRI was higher than SARI. In 2016 the TPRI site was 36˚C/20 ˚C and in 2017 it was 33˚C/19 ˚C. The high temperatures in 2016 mostly occurred between March and May.

### Productivity across environments

Overall environment (site x year) had a strong influence on all measures of productivity, including lablab grain yield, biomass, and maize yield (Fig 2, S2 Table). All lablab accessions

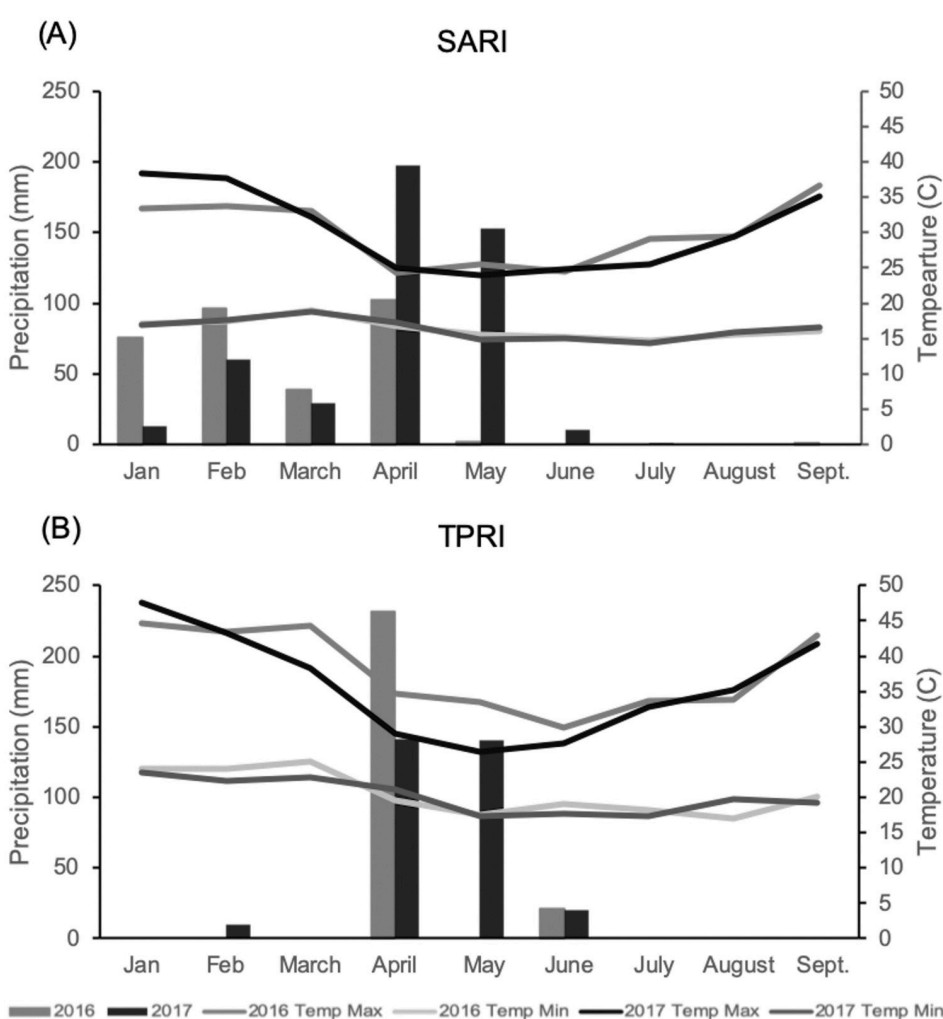

**Fig 1. Monthly precipitation and temperature for two growing season years across environments.** (A) SARI location (B) TPRI location.

produced low to nil grain yield at the TPRI site across both years, with averages of 31 kg ha[-1] in 2016 and 42 kg ha[-1] in 2017 (Fig 2c and 2d). Many late-flowering accessions did not set seed due to drought stress during reproductive stages. The highest grain yield in 2016 at TPRI was 116 kg ha[-1] produced by CIAT 22759 (#1) and in 2017 Q 6880B (#22) had the highest yield with 355 kg ha[-1]. In contrast, the SARI site had average grain yields at least ten times greater than TPRI, and a three-fold difference in yields between years, with 394 kg ha[-1] in 2016 and 1067 kg ha[-1] in 2017 (Fig 2a and 2b). Across accessions, differences in grain yield were seen at the SARI site with Karamoja Red (#17) having the highest grain yield in 2016 at 1001 kg ha[-1] and DL1002 (#4) had the highest grain yield in 2017 at 2029 kg ha[-1]. Grain yield did not differ under intercropped vs sole crop management across all environments (S2 Table). Grain yield of the cowpea reference species (#31) was higher than all lablab accessions at the TPRI site, with 224 kg ha[-1] in 2016 and 1071 kg ha[-1] in 2017. At the SARI site, cowpea out yielded all but one lablab accession in 2017, with a yield of 1864 kg ha[-1] whereas in 2016 it only yielded 395 kg ha[-1], which ranked it midway among lablab accessions for that same year. No evidence of cowpea grain yield reduction was found under intercrop vs sole crop management.

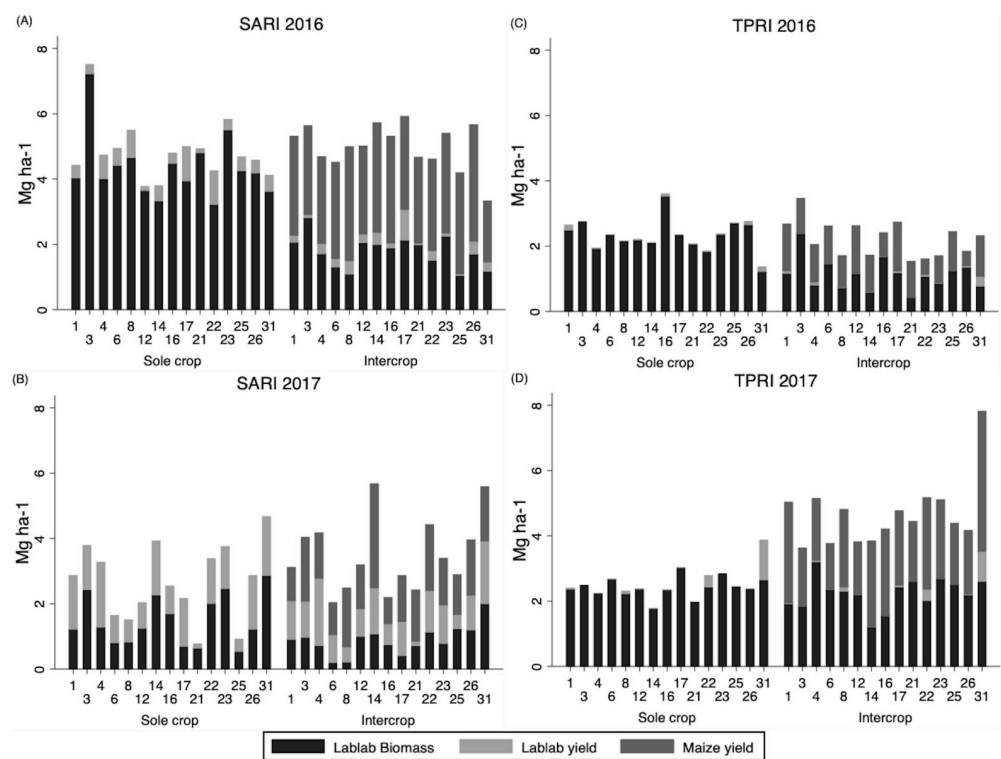

**Fig 2. Grain and biomass yield across the four environments sole planted and intercropped with maize.** (A) SARI 2016 (B) SARI 2017 (C) TPRI 2016 (D) TPRI 2017.

Biomass production differed across environments, accession, and by management (Fig 2, S2 Table). In contrast to grain yield, biomass produced at TPRI was comparable to that produced at SARI. Sole cropped lablab generally produced greater biomass than intercropped lablab, except for TPRI 2017 where the final maize population averaged only 20,200 ha$^{-1}$. Among sole cropped lablab, SARI 2016 had the highest biomass (4.4 Mg ha$^{-1}$), whereas SARI 2017 had the lowest (1.4 Mg ha$^{-1}$). Intercropped lablab produced the greatest biomass at TPRI 2017 (2.2 Mg ha$^{-1}$) and lowest at SARI 2017 (0.80 Mg ha$^{-1}$). Accessions of note for biomass production across environments include DL1001 (#3) which had the highest biomass of all accessions in SARI 2016 (5.02 Mg ha$^{-1}$), SARI 2017 (1.7 Mg ha$^{-1}$), and TPRI 2016 along with ILRI 6930 (2.6 Mg ha$^{-1}$). In contrast the top biomass producers at TPRI 2017 were Rongai (#23), Karamoja Red (#17), and DL1002 (#4) with 2.7–2.8 Mg ha$^{-1}$. The lowest biomass producers differed across environments, with Q 6880B (#22) the lowest (2.4 Mg ha$^{-1}$) in SARI 2016, Echo Cream (#6) the lowest (0.5 Mg ha$^{-1}$) in SARI 2017, PI 195851 (#21) the lowest (1.2 Mg ha$^{-1}$) in TPRI 2016, and ILRI 14437 (#14) the lowest (1.5 Mg ha$^{-1}$) in TPRI 2017.

Biomass for the cowpea reference crop followed similar trends to lablab biomass, with sole cropped cowpea generally producing greater biomass than intercropped cowpea (p = 0.0053; Fig 2). Sole cropped cowpea produced the greatest biomass in SARI 2016 (3.6 Mg ha$^{-1}$) and the least amount in TPRI 2016 (1.2 Mg ha$^{-1}$). Intercropped cowpea produced the same amount as sole cropped in TPRI 2016 (2.6 Mg ha$^{-1}$), but least amount in TPRI 2016 (0.8 Mg ha$^{-1}$).

Maize yield was also considered in assessing the productivity of intercropped lablab. Maize yield was not affected by accessions and only differed across environments (S2 Table). Maize yield was highest at SARI in 2016, with 3.0 Mg ha$^{-1}$ and lowest at TPRI in 2016 with

1.1 Mg ha$^{-1}$. Maize yields in 2017 were within this range, with 2.3 Mg ha$^{-1}$ at TPRI and 1.6 Mg ha$^{-1}$ at SARI.

Intercrop systems were overall more productive than sole cropped plots as shown by LER values greater than 1.7 across environments for both lablab grain LER and lablab biomass LER (S3 Table). An LER greater than 1 is indicative of an intercrop advantage over sole cropped production of the crops. Accessions of lablab performed in a highly similar manner, with no differences detected between accessions in terms of LER for either grain or biomass.

### Accession performance, stability, and environmental niches

Lablab accession performance across environments was ranked for grain yield and biomass production through the "Mean vs Stability" view of the GGE biplot (Fig 3). This view is based on mean performance and stability across environments within a mega-environment. The single arrowed axis is the average-environment coordination (AEC) abscissa and represents the average environment against which the accession performances are ranked. The arrow indicates the direction of higher mean performance and thus shows the rank of each accession. Stability of each lablab accession is represented by its location along the AEC ordinate (axis perpendicular to AEC abscissa), with the most stable accessions located on the AEC abscissa. The GGE biplots explained 98% of genotypic and genotype by environment variation across locations for grain yield performance and 79% of variation for biomass production (Fig 3). Accessions with above average grain yield in order of magnitude are DL1002 (#4), Karamoja Red (#17), Q 6880B (#22), ILRI 14437 (#14), CIAT 22759 (#1) and SARI Rongai (#26). Of these, DL1002 had the highest grain yield but was the most unstable as its rank was inconsistent across environments. Q 6880B was the most stable of the accessions that had above average grain yield (Fig 3A). Accession performance in relation to biomass production shows DL1001 (#3), Rongai (#23), and ILRI 6930 (#16) as having above average biomass yields, with Rongai also being the most stable (Fig 3B). In general, those accessions with above average grain yields were among the lowest in biomass production. No accessions had both above

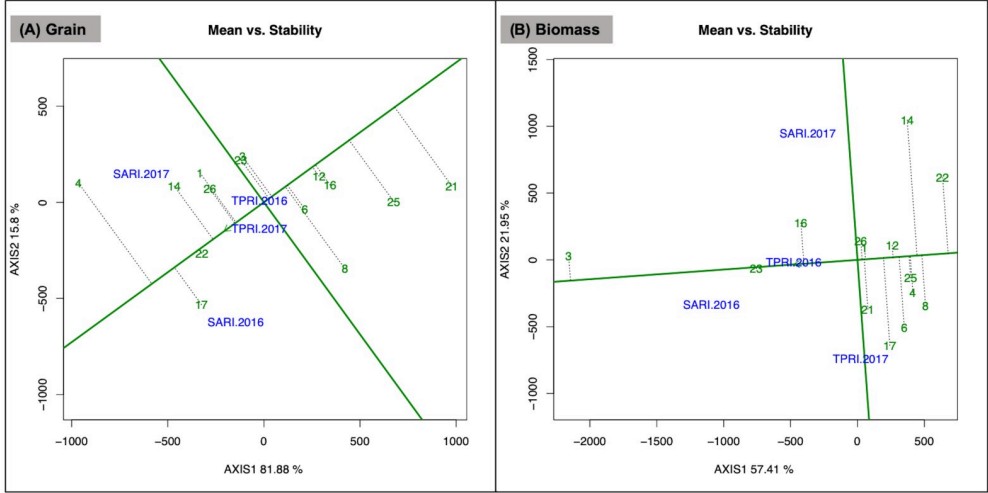

**Fig 3. Mean vs. stability view of GGE biplot for lablab subset accession grain and biomass performance.** (A) Grain yield and (B) biomass across the four test environments SARI 2016, SARI 2017, TPRI 2016, and TPRI 2017. The data were not scaled ("Scaling = 0"), environment centered ("Centering = 2"), and based on genotype-focused singular value partitioning ("SVP = 1").

average grain yield and biomass. Similarly, no accession had low stability in both grain yield and biomass.

The which-won-where view of the GGE biplot identifies the accessions which performed best in different environments as measured by grain yield and biomass (Fig 4). In this view, the lines originating from the biplot origin delineate sectors within which accessions and environments are matched as defined by their intersection with the polygon sides. The accession which performed best in each environment is the cultivar represented by the vertex of each sector. If all environments fall within a single sector, this indicates that a single accession did best across all environments. However, if environments fall in different sectors then different accessions performed best in different environments. In the which-won-where view for grain yield, SARI 2016 and SARI 2017 are identified as distinct environments within which different accessions performed well. DL1002 (#4) was the top performer in SARI 2017, with CIAT 22759 (#1), ILRI 14437 (#14), and SARI Rongai (#26) also best adapted to this environment for grain (Fig 4a). Karamoja Red (#17) was the best performer in SARI 2016 with Q 6880B (#22) also well adapted to this environment. TPRI 2016 and TPRI 2017 had low grain yields overall and were not clearly identified in a sector, suggesting that these environments are not well suited to lablab grain production. The remaining accessions did not clearly align to a test environment, which indicates that the environments in this study did not necessarily provide ideal conditions for grain production of these accessions.

In the which-won-where view for biomass, SARI 2016 and TPRI 2016 were identified as having similar accession performance, and thus were similar environments for biomass production (Fig 4b). Within these two environments DL1001 (#3) was the top biomass producer, with Rongai (#23) and ILRI 6930 (#16) performing well in these environments as well. TPRI 2017 was identified as a unique environment for biomass production within which Karamoja Red (#17) did best. Echo Cream (#6), PI 195851 (#21), Highworth (#8), DL1002 (#4) also did well in this environment. SARI 2017 did not align to a sector, suggesting it was not a suitable environment for maximizing biomass production. The remaining accessions that fell in different sectors without a clear environment signal are consistent with study environments as being not well suited to high biomass production for these accessions.

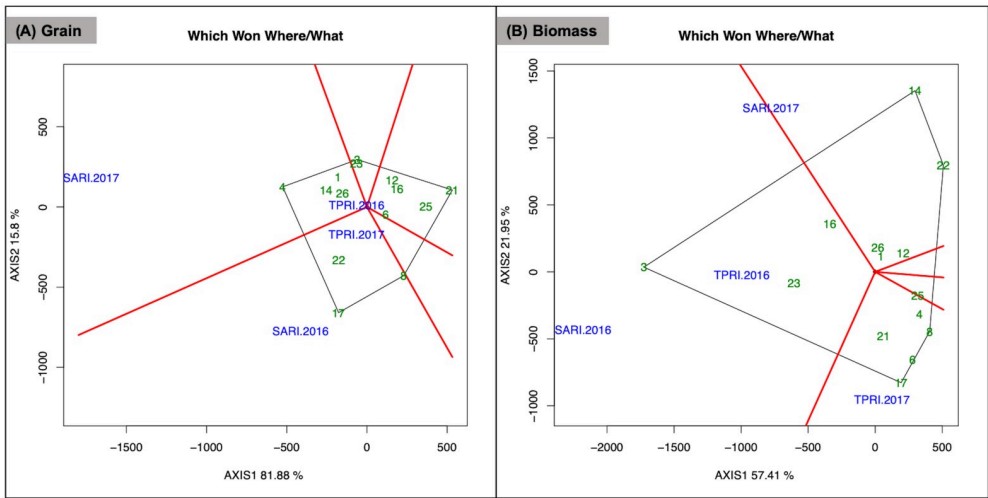

**Fig 4. Which-won-where view of GGE biplot for lablab subset accession grain and biomass performance.** (A) Grain yield and (B) biomass across the four test environments SARI 2016, SARI 2017, TPRI 2016, and TPRI 2017. The data were not scaled ("Scaling = 0"), environment centered ("Centering = 2"), and based on environment-focused singular value partitioning ("SVP = 2").

**Table 2. Factor loading and percentage of total variability explained for 2 factors in PCA using SARI 2017 data.**

|  | PC1 | PC2 |
|---|---|---|
| Eigenvalue | 2.689 | 1.499 |
| Variability (%) | 33.6 | 18.7 |
| Grain yield | 0.434 | 0.160 |
| Biomass | 0.473 | -0.160 |
| Soil nitrate | 0.104 | 0.319 |
| Nodule weight | 0.127 | 0.584 |
| $\delta^{15}N$ | 0.059 | -0.534 |
| Maturity | -0.408 | 0.435 |
| Plant pop | 0.417 | 0.082 |
| % N | -0.466 | -0.157 |

## Principal components analysis of productivity, growth, and nitrogen variables

In order to understand the relationship between grain yield, biomass, and nitrogen components across accessions, a PCA was performed on data from SARI 2017 where detailed nitrogen measurements were taken, including natural abundance assessment of biological N fixation. The variables of interest included lablab grain yield, biomass, soil nitrate, nodule weight, $\delta^{15}N$, maturity (days to 50% flowering), plant population and %N of biomass. Nodule weight and $\delta^{15}N$ values were used as a proxy for N fixation, with larger nodule weight assumed to be associated with increased N fixation and $\delta^{15}N$ values closer to zero associated with higher N fixation given that $\delta^{15}N$ signature of atmospheric $N_2$ is defined as zero [3,17].

The correlation matrix showed that grain yield was positively correlated with biomass ($r = 0.387$; $p<0.05$) and %N negatively correlated with grain ($r = -0.518$; $p<0.001$) and biomass ($r = -0.647$; $p<0.001$) (S4 Table). The variables were grouped into two components with eigenvalues greater than 1 and which explained 52.3% of the total variability among the variables (Table 2). The first component accounted for 33.6% of the variability and represented plant production as it was dominated by large loadings by grain yield, biomass, plant population and negatively with %N. The second component accounted for 18.7% of the variability and was associated with the nitrogen fixation variables nodule weight and $\delta^{15}N$ (negatively correlated) and soil nitrate (Table 2).

Biplots of the first two components with the variable loadings shows the distribution of accessions and block across productivity/growth (PC1) and N fixation (PC2) (Fig 5).

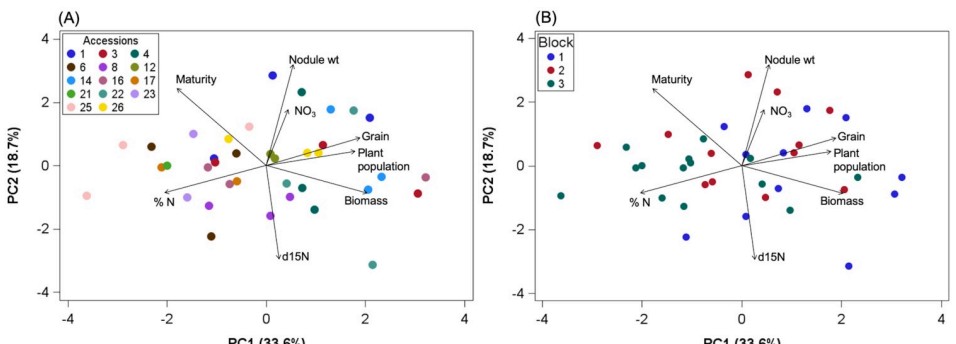

**Fig 5. Principal components analysis (PCA) biplot of SARI 2017 accession performance plotted against first two components with variable loading vectors (correlations between variables and PCs).** (A) PCA grouped by accession. (B) PCA grouped by block. Accumulated variability 52.3%.

Multivariate accession and block effects were found for PC1 but not PC2 (S5 Table). ILRI 14437 (#14) was found to have the highest productivity whereas SARI Nyeupe (#25) had the lowest. This suggests some accessions may be able to maintain high growth (yield, biomass) and N fixation, but for others N fixation may come at a cost to low growth.

## Discussion

### Lablab grain productivity by environment

Over the range of precipitation observed in the four environments, no clear trends emerged for lablab grain production. Overall, SARI 2017 had the highest average grain yields (1067 kg ha$^{-1}$) and was the environment with the highest seasonal precipitation (463 mm). This site year also had the highest grain yield of any accession, DL1002 with 2029 kg ha$^{-1}$, which was more than double the highest grain yield measured in 2016 at this site. SARI 2016 and TPRI 2017 had similar precipitation amounts (310–315 mm), but drastically different grain yields: 93–1001 kg ha$^{-1}$ at SARI, and 0–355 kg ha$^{-1}$ at TPRI. Previous studies also report a wide range in lablab yields with few consistent responses to precipitation. Whitbread et al. [22] tested 33 lablab accessions in South Africa and found in one site year at 475 mm of precipitation yields ranged from 1–576 kg ha$^{-1}$. Sennhenn et al. [23] tested lablab over a moisture gradient and found lablab grain yields as high as 1271 kg ha$^{-1}$ with 190 mm of rainfall. Our results are generally consistent with these studies as lablab yield ranged widely and was often higher than 500 kg ha$^{-1}$ under low precipitation (<500 mm), suggesting that drought-tolerance is a common trait in lablab.

We further found evidence for an interaction of temperature and precipitation in lablab grain production. The TPRI site in our study had between 252–315 mm of rainfall but was not suitable for grain production. Given that a similar amount of rainfall was seen at SARI in 2016 but with yields upwards of 1000 kg ha$^{-1}$, hot temperatures at the TPRI site seems to be the limiting factor for grain production. The TPRI site had both higher minimum and maximum temperatures than SARI, with maximum temperatures averaging 36°C in 2016 and 33°C in 2017. This was also hotter than the sites tested in Whitbread et al. [22](maximum 32°C), and in Sennhenn et al. [23] (maximum 31°C). A growth chamber experiment by Sennhenn et al. [24] testing the effect of temperature on development of lablab found that flowering was delayed at temperatures higher than 28°C. Our results support this finding and suggest that if grain yield is a priority, environments with maximum temperatures >33°C may not be suitable for lablab cultivation.

A third environment effect, intercrop versus sole crop management, was not found to affect grain yield (Fig 2, S2 Table). Previous descriptions of lablab suggest average grain yields around 1500 kg ha$^{-1}$ when sole cropped, but only 450 kg ha$^{-1}$ when intercropped [25]. Interestingly, in the highest grain yield environment, SARI 2017, average intercropped grain yields were higher (980 kg ha$^{-1}$) than these previous reports. In our study environments grain yields of sole cropping on average were generally modest, and plant densities after emergence were low (<50% emergence in SARI 2017) which may have supported minimal competition with maize in the intercrop and limited yield potential in the sole crop system.

There is a broad literature on grain legume intercropping with maize, and often lower legume grain yields are observed in intercrop vs sole cropped systems [26,27]. This may reflect farmer priorities as legume species are often grown as a secondary crop in an intercrop with maize, with low planting densities used by farmers. In some systems, such as common bean in Rwanda, bean production is the primary crop, and thus is planted at a higher density in the intercrop [26]. In most farmer systems, however, maize is the main crop and planting densities and spacing arrangements within and between rows are such that neither legume

grain nor maize yields is diminished[27]. Overall our findings did not detect differences in grain yield between sole and intercropped lablab accessions, which supports use of a simplified management system (either sole or intercropped) for future assessments.

## Lablab biomass production by environment

Biomass production by rainfall gradient demonstrated unclear trends. While many studies show biomass production increases with increased water availability, Sennhenn et al. [23] found that the increase in lablab biomass with increased water amounts was gradual in an irrigation gradient. In our study, the highest rainfall environment, SARI 2017 with 463 mm rainfall, had the lowest biomass production overall (1.1 Mg ha$^{-1}$). In contrast, SARI 2016 (315 mm) had the highest biomass overall (3.1 Mg ha$^{-1}$) but TPRI 2017 (311 mm), biomass was substantially lower (2.3 Mg ha$^{-1}$). Disease prevalence amongst legumes is also well known, and increased moisture may increase the severity of disease, thus negatively affecting biomass production [28].

Biomass production was less affected by high temperatures than grain in our study. The hot TPRI site, with maximum temperatures ranging from 33–36˚C, had average biomass yields of 1.8–2.3 Mg ha$^{-1}$ across the two years. Previous studies of lablab fodder production in semi-arid environments in east Africa have found similar biomass yields, with Sennhenn et al. [23] reporting 1.2–2.4 Mg ha$^{-1}$ and Macharia et al. [29] finding 2.5 Mg ha$^{-1}$ in eastern Kenya.

Biomass, in contrast to grain, was affected by intercrop management, with all environments having lower average biomass in intercrop versus sole crop systems, except in TPRI 2017. Across environments, sole cropped biomass ranged from 4.4 Mg ha$^{-1}$ (SARI 2016) to 1.4 Mg ha$^{-1}$ (SARI 2017). Intercropped biomass ranged from 2.2 Mg ha$^{-1}$ (TPRI 2017) to 0.8 Mg ha$^{-1}$ (SARI 2017). Despite the reduced biomass amounts produced in some intercrops, LERs based on lablab biomass ranged from 1.59 to 2.56. This is consistent with a strong production advantage for a forage lablab/maize intercrop system, particularly advantageous to farmers with livestock and limited land. Previous studies of lablab as a forage crop intercropped with maize have focused on lablab's potential as a dairy feed, where lablab was shown to have high potential as a high-quality forage. Maasdorp and Titterton [30] tested 15 legume crops in Zimbabwe with a range of growth habits for suitability as dairy feed in a maize intercrop system and found lablab's viny growth type to be complementary with maize with no or modest suppression of maize yield. This study found lablab biomass to be reduced in a maize intercrop, relative to sole lablab, but forage biomass produced was still higher than other legumes tested. Overall biomass in the intercrop was substantial, highlighting lablab's high potential in supplementing maize cropping systems [30]. Armstrong et al. [31] tested lablab's potential as a maize intercrop produced dairy feed in a cool temperate region and assessed nutritional properties, e.g., crude protein content. Of the legumes tested, lablab was the most acceptable within a dairy system because of the increased nutritional value it added to the system without suppressing maize yield [31]. While lablab may not have the highest production potential compared to other forage crops [32], it's multi-purpose qualities enhance its attractiveness to smallholder farmers interested in dual production of grain and forage [5]. Of note is the similar lablab biomass yields produced under intercrop and sole crop management in the hot environment of TPRI 2017, consistent with this crop as a dual use performer.

## Tradeoffs between grain and biomass

One key result from our study is the contrasting trends among accessions for biomass and grain yield as top biomass producers did not have high grain yield. Similarly, Ewansiha et al. [33] in assessing forty-six lablab accessions across two growing seasons also found an inverse

relationship between biomass and seed yield. Amongst another multipurpose legume, cowpea, Kabululu et al. [34] addressed this tradeoff in biomass and grain yield by testing determinate and indeterminate cultivars in mixtures to assess overall production. The authors found that while not all indeterminate/determinate mixtures outperformed monocultures, some mixtures were able to produce both high leaf and grain amounts. Such an approach has not been taken with lablab accessions, despite lablab and cowpea having comparable production qualities and range of growth habits. Interest in growing accessions in mixtures is further supported by a meta-analysis of cultivar mixtures and yield stability where mixtures often over-yielded relative to monocultures and this increase was more pronounced when mixtures included diverse traits [35]. This points to the value of identifying dual-purpose accessions in lablab, as growing a mixture of growth habits with complementary traits across cultivars may be necessary to meet farmer's multiple objectives.

### Genotype by environment interaction

Results observed for individual accessions indicated high plasticity, with accession performance varying by environment. This is consistent with many test environments being desirable to identify ideal environments for dual-purpose legumes. Generally in cultivar assessment the presence of genotype by environment interactions necessitates multiple test environments to identify suitable varieties for various production areas [36]. In assessing test environments for common bean across Africa using a GGE Biplot analysis, Kang et al. [8] identified redundant test environments for bean cultivars with implications for regional breeding centers. While an excessive use of test environments may be possible with a heavily studied crop such as common bean, dual-purpose legumes with a diverse genetic background such as lablab may well require many test environments [37]. This is supported by the GGE biplots of lablab accessions included in this study (Fig 4), where nine accessions for grain and six for biomass did not clearly align with the test environments, suggesting that further environments are needed to identify suitable growing niches for grain and biomass production, in addition to areas that are suitable for both.

An initial step towards identifying suitable environmental niches for lablab by mapping maximum temperature thresholds across Tanzania shows that the niche for high lablab biomass performance is substantially larger than it is for grain yield (Fig 6). Furthermore, these areas have substantial overlap with high livestock production areas. Future lablab performance studies in Tanzania should focus on these areas of overlap between high livestock and hot environments to expand the test environments used for lablab and thus gain additional insight on lablab accessions' environmental parameters.

This figure was produced by AN using QGIS version 2.18.2 and the following public domain data sources: Wan et al. [18] 8-day land surface temperature/emissivity; Arino et al. [38]land cover map (GlobCover); HarvestChoice [39] livestock prevalence (TLU).

### Individual accession performance

Across the four environments studied here, six of the 14 lablab accessions had above average grain yield and four of these accessions (DL1002, ILRI 14437, CIAT 22759, SARI Rongai) performed best under SARI 2017 growing conditions. The other top grain producers, Karamoja Red and Q 6880B, were best in SARI 2016. Of these accessions, three (DL1002, Q 6880B, ILRI 14437) have been identified in other studies as also having high production potential, especially as a short season legume crop [15, 22, 23, 40]. Q 6880B in particular was found by Sennhenn et al. [24] to be photoperiod insensitive even in higher temperatures, which could explain how Q 6880B was one of the few accessions to produce grain at the hot TPRI site and

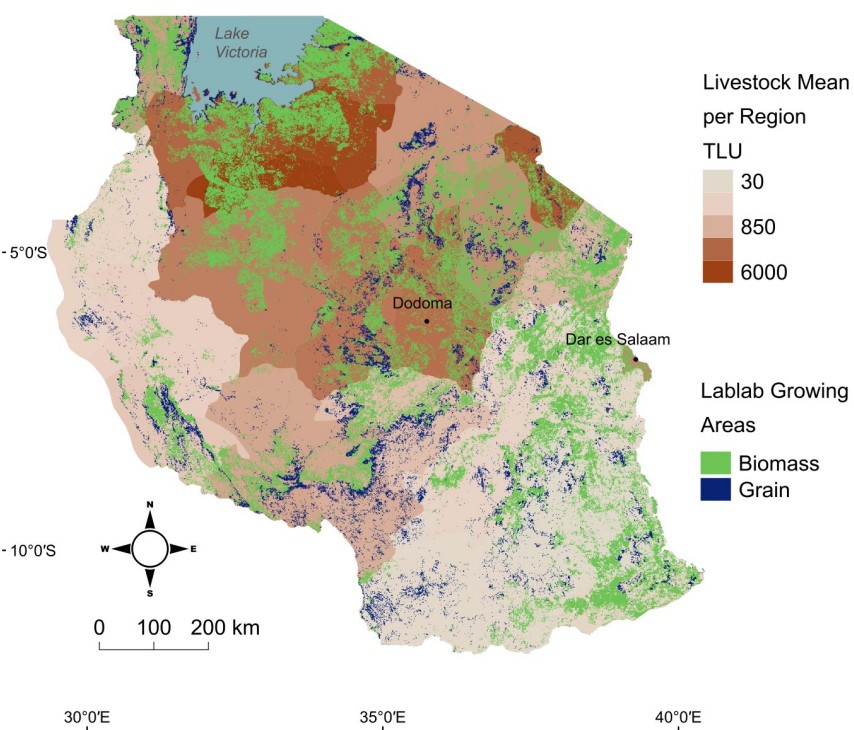

**Fig 6. Lablab potential growing areas and average regional livestock populations reported in Tropical Livestock Units (TLU) for mainland Tanzania.** Lablab potential based on maximum temperature thresholds for optimal grain (28˚C) and biomass (40 ˚C) production from December—August (10-year average) masked to agricultural land. Optimal grain areas are also suitable for biomass, but biomass areas are unsuitable for grain production. Livestock populations calculated as mean pixel TLU per region.

suggests that this accession may be best suited for promotion as a heat-tolerant grain variety. In our study, only three of the lablab accessions (DL1001, Rongai, ILRI 6930) had above average biomass yields (Fig 3B), and these accessions were best suited to both SARI 2016 and TPRI 2016 environments. The Rongai accession is a common lablab variety used for forage production, and previous studies have also noted it's high forage potential [31,37]. Interestingly, another lablab accession often promoted for forage, Highworth, was not found to be a top biomass producer in our study environments [25].

Of the accessions included in this study, four (DL1002, Echo Cream, Karamoja Red, Q 6880B) were chosen partway through the study for continuation in on-farm trials with the goal of identifying accessions for registration in Tanzania. Of these four, all except for Echo Cream were top grain producers. None of these accessions however were top performers in biomass, reflecting a preference for grain production in promoting lablab in northern Tanzania. Additionally, all accessions chosen for continuation are early-mid maturity types, a common preference in crop breeding programs [41]. Snapp et al. [42] however highlight the risk in a narrow selection of short-statured, early maturity crop types, including perpetuating an unsustainable simplified agricultural production system. Diverse crop growth types with dual purpose traits provide options for crop livestock integration and soil fertility enhancement, suggesting that there are risks associated with reductions in crop diversity through selection for a narrow range of traits. For example, lablab accessions with high biomass production in hot environments may be desirable for farmers in these locations, especially given widespread livestock husbandry. Farmers require expanded crop options, and a wide range of lablab accessions could help address these needs [14].

## Lablab performance tradeoffs

The SARI 2017 sub-study provided the first systematic assessment in lablab that we know of to quantify variability in accession N fixation traits, biomass, grain yield and soil N status. In this environment, tradeoffs were modest between biomass and grain, with some accessions identified that had high productivity and similar N fixation as those with low productivity. In the only other study of lablab genetic variation in N fixation, Ewansiha et al. [33], note that late maturing lablab varieties generally were associated with copious growth and large amounts of accumulated biologically fixed N, yet generally had low nodulation compared to earlier accessions. Our study found a similar trend in nodulation as those with higher nodule weight and low d15N (suggesting higher N-fixation) were early-mid maturing accessions. However, previous studies that estimated N fixation rates in lablab report percentages from 35–89% [32,43,44], suggesting that total N amounts in biomass might not imply greater amounts of $N_2$, as Ewansiha et al. indicate especially if N fixation rate differences are due to accession type. Further study is needed to assess lablab N fixation potential across accession types and to understand the relationship between maturity type and N-fixation [33], with clear implications for sustainability of multi-purpose legumes in smallholder farming systems.

## Lablab potential in smallholder farming systems

Lablab accessions provide unique options that address the multiple needs of farmers who are managing complex cropping systems, with clear potential to expand dual use legume production in hot environments across Tanzania (Fig 6). This study provides a methodology for identifying lablab accessions suitable to current farming systems, with the potential to improve overall sustainability. Accessions were identified as high performers in terms of grain or biomass, with particularly strong forage biomass performers identified for hot, dry environments, which could be introduced to support sustainable intensified livestock production in Tanzania [45]. At the same time, high environmental plasticity was observed for dual use strong performers, consistent with the need for broader environmental testing of accessions for dual use. Further, the study provided evidence that incorporating lablab into maize cropping systems as an intercrop would allow farmers to achieve sufficient grain and forage yield without having to commit land solely to lablab. The maize-lablab system was also suitable for accession screening, providing consistent results to sole lablab under hot dry environments. A recommendation coming out of our study is that lablab production and accession evaluations be conducted using intercrop rather than sole conditions, as this is most applicable to small-scale farming systems.

## Conclusion

While common bean is the most widely grown grain legume in Tanzania, its production area is limited by temperature and precipitation, thus limiting current legume production [46,47]. Expanding the temperature and rainfall range in which legumes are produced would therefore increase Tanzania's legume production area. Lablab accessions in our study produced substantial amounts of grain and biomass in hot, dry environments that were 6˚C above common bean's 25˚C max temperature threshold [47]. In a review of lablab's genetic diversity and value as a multi-purpose crop, Maass et al. [14] note lablab's greater drought tolerance over common bean and cowpea. While our study in a low rainfall, high heat environment showed cowpea to have greater grain production than lablab, lablab's high grain market value, particularly in northern Tanzania, and its ability to retain high-quality forage much longer than cowpea makes it a highly desirable drought tolerant crop. Further, farmer demand in hot, dry areas with high livestock dependency may be towards a drought tolerant forage legume such as lablab.

Dual use traits in crops is an under studied area of research and could provide key insights when integrated into methodology to assess novel legume crops for suitability of fit to cropping systems. The approach described here is a systematic means to evaluate lablab accessions by environment, that considers contributions to sustainability, as well as productivity, to expand crop options on African smallholder farms.

## Supporting information

**S1 Table. Soil properties of study site.** Numbers in parentheses standard errors.
(DOCX)

**S2 Table. Type 3 ANOVA of lablab grain yield, biomass, and maize yield.**
(DOCX)

**S3 Table. LER for lablab.**
(DOCX)

**S4 Table. Correlation matrix from PCA of SARI 2017 data.** Measurements from sole cropped plots only.
(DOCX)

**S5 Table. Type 3 ANOVA of PC1 and PC2 from PCA of SARI 2017 data.**
(DOCX)

**S6 Table. Characterization of nodules sampled at SARI 2017 site.** Nodules sampled in sole cropped plots only.
(DOCX)

**S7 Table. Soil nitrate from 0–20 cm depth.**
(DOCX)

## Acknowledgments

The authors thank the researchers and field staff at the Tanzania Agricultural Research Institute (TARI) Selian Centre for their extensive support in data collection.

## Author Contributions

**Conceptualization:** Alison Nord, Neil R. Miller, Wilfred Mariki, Laurie Drinkwater, Sieglinde Snapp.

**Data curation:** Alison Nord, Neil R. Miller, Wilfred Mariki.

**Formal analysis:** Alison Nord, Laurie Drinkwater, Sieglinde Snapp.

**Funding acquisition:** Sieglinde Snapp.

**Methodology:** Alison Nord, Neil R. Miller, Wilfred Mariki, Laurie Drinkwater, Sieglinde Snapp.

**Project administration:** Neil R. Miller, Wilfred Mariki.

**Supervision:** Neil R. Miller, Wilfred Mariki, Sieglinde Snapp.

**Writing – original draft:** Alison Nord.

**Writing – review & editing:** Alison Nord, Neil R. Miller, Laurie Drinkwater, Sieglinde Snapp.

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
