## [Decision Letter · Decision Letter 0]

31 Oct 2019

PONE-D-19-27128

Investigating the diverse potential of a multi-purpose legume, Lablab purpureus (L.) Sweet, for smallholder production in East Africa

PLOS ONE

Dear Dr. Nord,

Thank you for submitting your manuscript to PLOS ONE. After careful consideration, we feel that it has merit but does not fully meet PLOS ONE’s publication criteria as it currently stands. Therefore, we invite you to submit a revised version of the manuscript that addresses the points raised during the review process.

We would appreciate receiving your revised manuscript within 90 days. To enhance the reproducibility of your results, we recommend that if applicable you deposit your laboratory protocols in protocols.io, where a protocol can be assigned its own identifier (DOI) such that it can be cited independently in the future. For instructions see: http://journals.plos.org/plosone/s/submission-guidelines#loc-laboratory-protocols

We look forward to receiving your revised manuscript.

Kind regards,

P. Pardha-Saradhi, Ph.D.

Academic Editor

PLOS ONE

Journal Requirements:

2. We note that Figure [6] in your submission contains a map image which may be copyrighted. All PLOS content is published under the Creative Commons Attribution License (CC BY 4.0), which means that the manuscript, images, and Supporting Information files will be freely available online, and any third party is permitted to access, download, copy, distribute, and use these materials in any way, even commercially, with proper attribution. For these reasons, we cannot publish previously copyrighted maps or satellite images created using proprietary data, such as Google software (Google Maps, Street View, and Earth). For more information, see our copyright guidelines: http://journals.plos.org/plosone/s/licenses-and-copyright.

You may seek permission from the original copyright holder of Figure(s) [6] to publish the content specifically under the CC BY 4.0 license. 

If you are unable to obtain permission from the original copyright holder to publish these figures under the CC BY 4.0 license or if the copyright holder’s requirements are incompatible with the CC BY 4.0 license, please either i) remove the figure or ii) supply a replacement figure that complies with the CC BY 4.0 license. Please check copyright information on all replacement figures and update the figure caption with source information. If applicable, please specify in the figure caption text when a figure is similar but not identical to the original image and is therefore for illustrative purposes only.

Reviewers' comments:

Reviewer's Responses to Questions

**Comments to the Author**

1. Is the manuscript technically sound, and do the data support the conclusions?

Reviewer #1: Yes

2. Has the statistical analysis been performed appropriately and rigorously? 

Reviewer #1: Yes

3. Have the authors made all data underlying the findings in their manuscript fully available?

Reviewer #1: Yes

4. Is the manuscript presented in an intelligible fashion and written in standard English?

Reviewer #1: Yes

5. Review Comments to the Author

Reviewer #1: The elaborate research conducted with diverse cultivars of under exploited crop- Lablab purpureus at two locations with two system approach generated vast data sets. The experiments laid out and observations recorded are sufficient to conclude the results. The appropriate analysis of data enabled to identify the cultivar for both selected sites and also as an inter crop. However the following points may be considered for review the results

- The duration of legumes impact their tolerance to abiotic stresses as selected cultivars are with early to very late duration. Hence for assessing their suitability one has to compare similar duration cultivars.

- Both edaphic and weather conditions varied in selected two sites of experimentation. The information on one site soil conditions was provided. A comparison need to be done on impact of soil conditions while assessing the performance of cultivars.

- Too many parameters were included for identifying suitable cultivar/s for biomass and seed yield. Nitrogen component needs to be addressed separately and some more detailed experimentation is required, hence this parameter may be omitted.

- Many statements are repetitive and lengthy such as explanation of results for two sites for two years may be made as single statement

6. PLOS authors have the option to publish the peer review history of their article (what does this mean?). If published, this will include your full peer review and any attached files.

Reviewer #1: Yes: Dr MADDI VANAJA

---

## [Author Response · Author response to Decision Letter 0]

21 Dec 2019

Dear PLOS ONE Editor,

Below are our responses to the comments provided by the reviewer and academic editor. 

We have revised the submission to address all comments, editor and reviewer. In response to the journal requirements highlighted by the academic editor, we have edited the manuscript to comply with PLOS ONE’s style requirements. Concerning Figure 6 which was highlighted by the editor as possibly being copyright material, we have clarified in the caption that we produced this map, so it is not copyrighted. We added information for clarification in the figure heading stating that the map was produced by us using QGIS software and public domain data, which does not impose any restrictions on maps produced and therefore is not copyrighted. Additionally, captions for supporting information files have been added at the end of the manuscript as requested (on page 35). 

We trust these revisions address the edits suggested by the editor and reviewer.

Sincerely,

Alison Nord

Response to Reviewer #1:

The duration of legumes impact their tolerance to abiotic stresses as selected cultivars are with early to very late duration. Hence for assessing their suitability one has to compare similar duration cultivars.

Response: We appreciate this point by the reviewer but would like to clarify that within this study we are looking at variation both across and within growth types, and therefore used the comprehensive statistical model shown on page 10 to assess cultivar performance.

Both edaphic and weather conditions varied in selected two sites of experimentation. The information on one site soil conditions was provided. A comparison need to be done on impact of soil conditions while assessing the performance of cultivars.

Response: We have provided the soil conditions as supplementary information as soil conditions of our study sites were not an explicit factor in the analysis (as only two sites were represented).

Too many parameters were included for identifying suitable cultivar/s for biomass and seed yield. Nitrogen component needs to be addressed separately and some more detailed experimentation is required, hence this parameter may be omitted.

Response: We have added clarification in the methods section on page 9 to highlight that the PCA performed for one site year was part of data collected as a sub study on nitrogen and productivity. This specific analysis looked at variability across accessions for parameters related to nitrogen fixation and productivity, not the overall suitability of cultivars for biomass and grain yield which was addressed in the larger study. As such, we feel it is important to include this sub study as an initial step towards understanding N dynamics across lablab cultivars for which research is currently lacking in the literature.

Many statements are repetitive and lengthy such as explanation of results for two sites for two years may be made as single statement

Response: We have addressed this concern and have made efforts throughout the manuscript to condense statements, as seen through the manuscript track changes. See especially page 14 of the results.

---

## [Editor Report · Decision Letter 1]

30 Dec 2019

Investigating the diverse potential of a multi-purpose legume, Lablab purpureus (L.) Sweet, for smallholder production in East Africa

PONE-D-19-27128R1

Dear Dr. Nord,

We are pleased to inform you that your manuscript has been judged scientifically suitable for publication and will be formally accepted for publication once it complies with all outstanding technical requirements.

With kind regards,

P. Pardha-Saradhi, Ph.D.

Academic Editor

PLOS ONE
---

## [Editor Report · Acceptance letter]

6 Jan 2020

PONE-D-19-27128R1 

Investigating the diverse potential of a multi-purpose legume, *Lablab purpureus* (L.) Sweet, for smallholder production in East Africa 

Dear Dr. Nord:

I am pleased to inform you that your manuscript has been deemed suitable for publication in PLOS ONE. Congratulations! Your manuscript is now with our production department. 

With kind regards,

on behalf of

Prof. P. Pardha-Saradhi 

Academic Editor

PLOS ONE